# Synergistic Drug Combinations Prevent Resistance in ALK+ Anaplastic Large Cell Lymphoma

**DOI:** 10.3390/cancers13174422

**Published:** 2021-09-01

**Authors:** Giulia Arosio, Geeta G. Sharma, Matteo Villa, Mario Mauri, Ilaria Crespiatico, Diletta Fontana, Chiara Manfroni, Cristina Mastini, Marina Zappa, Vera Magistroni, Monica Ceccon, Sara Redaelli, Luca Massimino, Anna Garbin, Federica Lovisa, Lara Mussolin, Rocco Piazza, Carlo Gambacorti-Passerini, Luca Mologni

**Affiliations:** 1Department Medicine and Surgery, University of Milano-Bicocca, 20900 Monza, Italy; g.arosio17@campus.unimib.it (G.A.); geeta.geeta@unimib.it (G.G.S.); m.villa96@campus.unimib.it (M.V.); mario.mauri@unimib.it (M.M.); ilaria.crespiatico@unimib.it (I.C.); diletta.fontana@unimib.it (D.F.); chiara.manfroni@unimib.it (C.M.); cristina.mastini@unimib.it (C.M.); marina.zappa.mz@gmail.com (M.Z.); vera.magistroni@unimib.it (V.M.); monica.ceccon@unimib.it (M.C.); sara.redaelli@unimib.it (S.R.); admin@lucamassimino.com (L.M.); rocco.piazza@unimib.it (R.P.); carlo.gambacorti@unimib.it (C.G.-P.); 2Department Hematology & Hematopoietic Cell Transplantation, City of Hope National Medical Center, 1500 E Duarte Rd, Duarte, CA 91010, USA; 3Department Gastroenterology, Humanitas University, Pieve Emanuele, 20090 Milano, Italy; 4Department Women’s and Children’s Health, Clinic of Pediatric Hemato-Oncology, University of Padua, 35122 Padova, Italy; anna.garbin.1@studenti.unipd.it (A.G.); federica.lovisa@unipd.it (F.L.); lara.mussolin@unipd.it (L.M.); 5Non-Hodgkin Lymphoma Unit, Istituto di Ricerca Pediatrica Fondazione Città della Speranza, 35122 Padova, Italy

**Keywords:** ALK, ALCL, drug combination, synergy, resistance

## Abstract

**Simple Summary:**

Despite success of targeted therapy, cancer cells very often find a way to survive treatment; this eventually causes a tumor to relapse. In a particular type of lymphoma carrying a specific chromosomal rearrangement named anaplastic large-cell lymphoma (ALCL), selective drugs targeting the cause of the disease can induce spectacular remission of chemotherapy-resistant cancer. However, the lymphoma relapses again in about half of the cases, leaving no therapeutic options. We studied the possibility to combine two simultaneous treatments in order to prevent the relapse, starting from the hypothesis that acquiring resistance to two drugs at the same time is statistically very unlikely. We demonstrate that treating lymphoma cells with drug combinations has superior efficacy in comparison with single drug treatments, both in cell cultures and in mice.

**Abstract:**

Anaplastic lymphoma kinase-positive (ALK+) anaplastic large-cell lymphoma (ALCL) is a subtype of non-Hodgkin lymphoma characterized by expression of the oncogenic NPM/ALK fusion protein. When resistant or relapsed to front-line chemotherapy, ALK+ ALCL prognosis is very poor. In these patients, the ALK inhibitor crizotinib achieves high response rates, however 30–40% of them develop further resistance to crizotinib monotherapy, indicating that new therapeutic approaches are needed in this population. We here investigated the efficacy of upfront rational drug combinations to prevent the rise of resistant ALCL, in vitro and in vivo. Different combinations of crizotinib with CHOP chemotherapy, decitabine and trametinib, or with second-generation ALK inhibitors, were investigated. We found that in most cases combined treatments completely suppressed the emergence of resistant cells and were more effective than single drugs in the long-term control of lymphoma cells expansion, by inducing deeper inhibition of oncogenic signaling and higher rates of apoptosis. Combinations showed strong synergism in different ALK-dependent cell lines and better tumor growth inhibition in mice. We propose that drug combinations that include an ALK inhibitor should be considered for first-line treatments in ALK+ ALCL.

## 1. Introduction

The emergence of resistance is the major limitation to cancer cures. Tumors are characterized by genetic instability that leads to random accumulation of mutations in different subclones, resulting in high clonal heterogeneity. The various subpopulations present at diagnosis respond differently to therapies. In this scenario, some tumor cells have a higher probability to survive and will eventually re-expand after a bottleneck induced by a treatment. In most cases, this makes the monotherapy approach rather ineffective in the long term [1]. However, the chance of a cell being resistant to two simultaneous treatments is much lower, suggesting that drug combinations may help avoid resistance. In theory, an upfront coadministration of treatments at first line may lead to the prevention (rather than treatment) of resistant disease. We hypothesize that rationally designed upfront polytherapies might improve the long-term outcome of patients. To investigate this, we applied a combined (targeted) therapy strategy to ALK+ ALCL, a non-Hodgkin peripheral T-cell lymphoma that depends on the activity of ALK fusion proteins, which support cancer growth by constant activation of JAK/STAT, RAS/MEK/ERK and PI3K/AKT pathways [2,3]. The ALCL99 treatment protocol has shown great activity for pediatric cases, while CHOP (cyclophosphamide, doxorubicine, vincristine, prednisone) and other cyclophosphamide-based regimens, such as BV-CHP or CHOEP, represent the standard therapy in adult ALCL [4,5]. Despite excellent prognosis, a significant proportion of patients relapse. In patients resistant or refractory to chemotherapy, the ALK inhibitor crizotinib has shown important therapeutic activity [6,7]. However, 30–40% of patients still develop resistance to crizotinib monotherapy, with ensuing poor prognosis. We previously described different mechanisms of resistance to ALK inhibitors, including ALK point mutations, gene amplification and ALK-independent bypass mechanisms [8,9,10]. Although clonal heterogeneity has not been thoroughly investigated in ALK+ ALCL, the emergence of various mutants under ALK therapy clearly suggests the pre-existence of different subclones within the starting bulk population. Second-generation ALK inhibitors have been developed to treat ALK-mutated crizotinib-resistant disease, often resulting in different sensitivity profiles of mutant clones toward the different compounds [3,11]. In theory, one could exploit these differences by combining ALK inhibitors with complementary activities on mutants, achieving a wider coverage of mutations that can be inhibited. For ALK-independent resistance, novel approaches are needed. The MAPK pathway has been involved in resistance to ALK inhibitors in several settings [9,12,13,14]. In particular, we previously described marked upregulation of ERK activity in lorlatinib-resistant tumors [9]. Furthermore, NPM/ALK has been shown to act through epigenetic regulation of gene expression and epigenetic changes have been observed in drug resistant ALCL samples [15,16,17,18,19]. Indeed, the DNA methyltransferase inhibitor 5-aza-20-deoxycytidine (decitabine) has shown antitumor activity in ALK+ ALCL [20]. Third, we recently found that ALK-independent resistance to crizotinib does not correlate with a response to chemotherapy, suggesting that a combination of crizotinib and chemotherapy could prevent relapse [21].

In this work, we tested the efficacy of crizotinib combinations with either standard cytotoxic chemotherapy, or the epigenetic drug decitabine, or the MEK1/2 inhibitor trametinib. In addition, second and third generation ALK inhibitors such as ceritinib, brigatinib and lorlatinib were combined with each other or with crizotinib.

## 2. Materials and Methods

### 2.1. Chemicals, Antibodies and Cells

4-Hydroperoxy-cyclophosphamide (4-HC, the active metabolite of cyclophosphamide), vincristine and doxorubicin, which compose the CHO treatment (2 µM 4-HC, 20 nM doxorubicin; 0.5 nM vincristine), were purchased from SIGMA-Aldrich (St. Louis, MO, USA). All other drugs were from Selleck Chem (Houston, TX, USA). For in vitro analyses, drugs were dissolved in DMSO; for in vivo studies, crizotinib and trametinib were prepared fresh in 0.5% carboxymethylcellulose/0.1% Tween 80; all the other compounds were dissolved in PBS. Primary antibodies (Appendix A) were diluted 1:1000 in 5% BSA and incubated overnight. Anti-mouse and anti-rabbit secondary antibodies (Bio-Rad, Hercules, CA, USA) were used 1:2000 in 5% BSA. ALK+ ALCL cell lines Karpas-299, SUP-M2 and SU-DHL-1, and ALK-negative myelomonocytic leukemia U937 cells were purchased from ATCC. Cells were cultured in RPMI-1640 medium (Euroclone, Siziano, Italy) supplemented with 10% OptiClone fetal bovine serum, 2 mM L-glutamine, 100 units/mL penicillin G, 80 μg/mL gentamicin and 20 mM HEPES, in a humidified atmosphere at 37 °C and 5% CO_2_.

### 2.2. Long-Term Cell Cultures

ALCL cells (10^7^ cells/condition) were kept in the presence of compounds for a maximum of 100 days. Cell growth was monitored by counting live cells three times a week using Trypan Blue. Medium and drugs were replenished every 72 h and cultures were maintained at a concentration of 3 × 10^5^ cells/mL to keep exponential growth. Cumulative cell number N at any time point n takes into account previous cell culture dilutions according to the formula: N(n) × d1 × d2 … × dn − 1 where d is dilution factors at times 1 to n − 1. Relative growth was obtained by dividing cumulative cell number by the starting cell number at day 0.

### 2.3. Proliferation Assays, Synergy and Statistical Analysis

Cells (10^4^/well) were seeded in 96-well plates and treated with single or constant-ratio combined drugs for 72 h. Proliferation was assessed using CellTiter 96^®^ AQueous One Solution Cell Proliferation Assay System (Promega, Madison, WI, USA) according to instructions. Dose–effect curves and combination index (CI) values were calculated using CalcuSyn software following the theory of Chou and Talalay [22] where CI<1, equal to 1 and >1 indicate synergism, additive effects and antagonism, respectively. Statistical differences between groups were analyzed by *t*-test; significance was defined by *p*-values (*, *p* ≤ 0.05; **, *p* ≤ 0.01; ***, *p* ≤ 0.001).

### 2.4. Cell Cycle and Apoptosis

Cell cycle was studied by flow cytometry after propidium iodide (PI) staining. Cells were treated with the indicated compounds, collected, washed in PBS and fixed in cold 70% ethanol for 30 min at 4 °C. Samples were then resuspended in PBS containing 1 mg/mL PI and 10 mg/mL RNase A for 30 min at 37 °C and analyzed using the Attune™ NxT Flow Cytometer (ThermoFisher Scientific, Waltham, MA, USA). To assess apoptosis, 10^6^ cells/condition were stained with Annexin V-FITC and PI according to eBioscence^TM^ Annexin V-FITC Apoptosis Detection Kit (ThermoFisher Scientific) protocol and analyzed by flow cytometry.

### 2.5. Soft Agar Colony Assay

Ten thousand cells/well were embedded in cell culture medium containing 0.33% low-melting agarose (Sigma, St. Louis, MO, USA) and either DMSO (0.1%) or the indicated drugs and seeded in triplicate on a bottom layer of 0.5% low-melting agarose in six-well plates, as described [23]. Colonies were counted after 21 days.

### 2.6. Western Blot Analysis

For immunoblotting analyses, cells were collected and lysed in Laemmli buffer supplemented with 10% β-mercaptoethanol. Lysates were denatured at 95 °C for 20 min, loaded on SDS-PAGE, transferred to a nitrocellulose membrane (GE Healthcare, Chicago, IL, USA) and incubated overnight at 4 °C with the indicated primary antibody (Appendix A). After incubation with horseradish peroxidase-conjugated secondary antibody, the ChemiDoc XRS+ System (Bio-Rad, Hercules, CA, USA) and Image Lab software (Bio-Rad) were used to detect and analyze protein bands.

### 2.7. Quantitative Real-Time PCR

Gene expression was assessed by quantitative real-time PCR using Brilliant III Ultra-Fast SYBR Green^®^ QPCR Master Mix (Agilent Technologies, Santa Clara, CA, USA); thermal cycling conditions were as follows: 3 min at 95 °C, followed by 35 cycles of 5 s at 95 °C and 20 s at 60 °C. Data were normalized to the expression level of the housekeeping gene glyceraldehyde 3-phosphate dehydrogenase (GAPDH). TaqMan Probe mixes (ThermoFisher) were used for the analysis of BIK (Hs00154189), BIM (BCL2L11; Hs00708019) and STAT5A (Hs00559637), with Luna^®^ Universal Probe qPCR Master Mix (New England Biolabs, Ipswich, MA, USA). Cycling conditions were set as follows: 10 s at 95 °C, followed by 40 cycles of 15 s at 95 °C and 1 min at 60 °C. The housekeeping gene β-glucuronidase (GUS) was used for normalization. Primers employed in this study were described previously [19,24] or are listed in Appendix A.

### 2.8. 3D Matrix-Embedded Cell Cultures

Three-dimensional (3D) cell cultures were set up using an ECM proteins rich matrix (Matrigel matrix-standard formulation, Corning, New York, NY, USA). In a 96-well plate pre-coated with Matrigel (5 mg/mL), cells were seeded at a concentration of 2.5 × 10^3^ cells/well, embedded in a layer of matrix diluted with RPMI medium (1:1) in a final volume of 60 µL. Plates were incubated at 37 °C for 1 h and then medium was added onto the top of the layer. After 4 days, 3D cultures were treated by adding single or combined drugs to the medium, which was replenished every 72 h. The growth of 3D cultures was monitored with ImageJ software by measuring the total area of the cell masses. 

### 2.9. In Vivo Experiments

Six-week-old female mice were purchased from Envigo (Milan, Italy) and kept under standard conditions following guidelines by the University of Milano-Bicocca ethical committee for animal welfare. The protocol was approved by the Italian Ministry of Health. Mice were injected subcutaneously with 10^7^ Karpas-299 cells in each flank; when tumors were measurable (average volume ≈ 200 mm^3^) mice were randomized to receive vehicle or single treatments or combinations, as follows: crizotinib (30 mg/kg) and trametinib (2 mg/kg) were administered by oral gavage once daily (q.d.); decitabine (0.8 mg/kg, q.d.) was given intraperitoneally (i.p.); mice in the chemotherapy arms received i.p. treatment every other day according to the following schedule: CHO (40 mg/kg cyclophosphamide, 3 mg/kg doxorubicin, 0.2 mg/kg vincristine) on days 1, 5, 9; cyclophosphamide alone, on days 3, 7, 11. Tumor volume was measured with a caliper using the formula: Volume (mm^3^) = d2 × D/2 where d is the shortest and D is the longest diameter of the tumor mass.

## 3. Results

### 3.1. Upfront Combined Treatments Prevent the Selection of Resistant Clones in ALK+ ALCL Cells

To investigate the feasibility of preventing drug resistance, ALCL cells were cultured in the presence of different single and combined drugs. The efficacy of two different approaches was evaluated. On the one hand, we evaluated the coadministration of crizotinib with non-specific anticancer treatments, such as decitabine or chemotherapy; the latter was represented by a combination of cyclophosphamide, doxorubicin and vincristine (CHO), mimicking a CHOP treatment [25]. On the other hand, combinations of two targeted therapies were tested, such as: (i) the ALK inhibitor crizotinib combined with a downstream signaling targeting drug (the MEK inhibitor, trametinib); (ii) two ALK inhibitors with different activity profiles against ALK mutants. In our in vitro model, the initial number of cells represents the heterogeneity of a tumor mass undergoing selective pressure and clonal evolution, and the uncontrolled expansion of the cell population reflects the failure of a treatment to prevent the onset of resistance and consequent tumor progression. To mimic clinical situations where the activity of drugs is suboptimal, leading to resistance and relapse, sub-lethal doses were employed, i.e., around IC50 or below. Preliminary tests were run to determine drug concentrations that would allow the outgrowth of resistant cells as single agents. Drugs were then used in combination using the same concentrations. We found that associating low dose crizotinib with CHO (Figure 1A–C), decitabine (Figure 1D–F) or trametinib (Figure 1G–I) completely suppressed the emergence of resistant cells. In all the trials, when two agents were combined, the number of lymphoma cells in culture was efficiently kept under control up to 100 days. By contrast, single treatments resulted in the selection of resistant clones in less than one month and caused the expansion of the tumor cell population. Down to one million starting cells allowed the outgrowth of a resistant clone, suggesting a frequency of at least 10^−6^ drug resistant (or drug tolerant) cells within the initial population. In one case (Karpas-299, crizotinib+decitabine) crizotinib dose could be further scaled down (50 nM) thanks to the high efficacy of the combination (Figure 1D).

These results were consistent not only among different combinations, but also across different cell lines, suggesting that these combinations are effective in NPM/ALK+ cells, independently of their genetic background. Furthermore, cells resistant to any monotherapy maintained sensitivity to the other drug of a combination, i.e., they did not show cross resistance in a standard 72-h growth assay, explaining the efficacy of dual targeting (Appendix A). Cells resistant to crizotinib did not carry mutations in the ALK kinase domain but expressed higher NPM/ALK transcript levels (Appendix A), which may explain their lower sensitivity to the inhibitor, as shown previously [8]. We then compared the efficacy of sequential vs. simultaneous treatments. In all three cases, sequential combinations failed: cells resistant to the first drug developed second-line resistance at some point (Appendix A), becoming resistant to both drugs. In contrast, simultaneous combinations confirmed their efficacy in this setting.

Combinations of two ALK inhibitors were based on the assumption that compounds with complementary (more distant) profiles of resistance would inhibit a wider spectrum of mutant subclones when used together (Appendix A) [26,27]. The inhibitors were combined at half the dose and compared to full-dose single treatments, since they hit the same target and compete with one another, at least on the wild-type enzyme. The majority of the tested combinations eventually allowed the expansion of resistant clones, and the only benefit was a delay in the development of resistance (Appendix A). Moreover, we could not identify a combination that would be effective in all cell lines, likely reflecting cell line-specific differences in the frequency of pre-existing mutant clones. Crizotinib+lorlatinib resulted to be efficacious in two out of three cell lines.

Altogether, these results suggest that a complete forestall of drug resistance can be achieved by combined treatments. However, blocking independent cellular relays appears to be more efficient than hitting a single target with two drugs.

### 3.2. Synergistic Interaction of Crizotinib with Decitabine, CHO and Trametinib

To investigate the reasons for the observed advantage of combinations over single treatments, drug interactions were characterized in short-term cultures. To this end, the treatments were combined at different constant ratios over a range of drug concentrations. The analysis of dose–effect curves allowed a quantitative measure of the efficacy of each combination, as indicated by combination index (CI) values (Table 1). In all three NPM/ALK+ cell lines, crizotinib combinations with chemotherapy (CHO; Figure 2A–C), decitabine (Figure 2D–F) and trametinib (Figure 2G–I) were found to have synergistic effects (CI < 1). Interestingly, while sensitivity to single agents may differ among the three cell lines under study, in all ALK+ cells the combined treatments caused significantly higher growth inhibition compared to single treatments. In contrast, no synergism was seen in ALK-negative cells, as they do not respond to ALK inhibitors (Table 1 and Appendix A). These results explain the efficacy of all these combined treatments in inhibiting ALCL cell growth in long-term cultures. On the other hand, the simultaneous administration of two ALK inhibitors resulted in additive or antagonistic interactions, as the effects of the combinations superimposed to, or were lower than, those of a single inhibitor (Appendix A). Since combining two ALK inhibitors did not improve the results achieved by single inhibitors, this type of combination was not explored further.

### 3.3. Combined Treatments Enhance Apoptosis in vitro and Tumor Growth Inhibition in Mice

The effects of combined treatments on the cell cycle were analyzed. Both single and combined drugs inhibited the progression of ALCL cells through the cell cycle, as shown by a contraction of the S phase compared to untreated cells. In addition, all the combinations induced a significant increase in the fraction of cells with sub-G1 DNA content compared to single treatments (Figure 3A). Annexin V staining confirmed higher cell death rate through induction of apoptosis by the combinations (Figure 3B). To further explore the effects on tumor progression and expansion, cells were seeded in soft agar in the presence of drugs. Almost full suppression of colony formation was associated with drug combinations which significantly enhanced the effect of single drugs (Figure 3C).

To confirm these data in a 3D microenvironment, we set up spheroid cultures of SUP-M2 embedded in an extracellular matrix. Treatments started when multicellular aggregates of approximately 10–50 cells were visible under a microscope. All three combinations resulted in nearly complete abrogation of spheroids growth, while single agents showed variable efficacy but eventually failed to block expansion (Figure 3D). Finally, to further validate the potentialities of this combined approach, we sought an in vivo proof of concept, employing a xenograft model of ALK+ ALCL. Combined therapy with crizotinib+decitabine or crizotinib+trametinib resulted in a significantly higher inhibition of tumor growth compared to single treatments, while tumors treated with crizotinib+CHO showed a trend toward better inhibition compared to single drugs (Figure 3E).

### 3.4. Molecular Effects of Combined Therapies

To assess the molecular mechanisms supporting an increased efficacy of combined treatments, quantitative PCR and Western blot analyses were performed on Karpas-299 and SUP-M2 cell lines treated with single or combined drugs. Crizotinib+CHO was found to upregulate transcription of pro-apoptotic genes BAK, BIK and BIM and of the CDKN1B gene, encoding for the cell cycle inhibitor p27Kip1 (Figure 4A–D). A summary of quantitative expression analyses of selected genes is reported in Appendix A. At protein level, expression of the anti-apoptotic protein BCL2 was reduced (Figure 4E). As cyclophosphamide is known to induce DNA strand breaks [28], phosphorylation of histone 2AX on serine 139 (γ-H2AX), a marker of DNA damage and cell death, was assessed. A marked increase in γ-H2AX signal was observed in the combination group, compared to control and single treatments (Figure 4E). As expected, CHO induced phosphorylated and total p53, but this effect was not further increased by the combination. The cell cycle inhibitor protein p27Kip1 was significantly upregulated by the combined treatment, in line with higher expression of the corresponding mRNA (CDKN1B) and with cell cycle arrest. In contrast, a peculiar effect was noted on p21Waf1/Cip1. Transcription of the CDKN1A gene increased, but the p21 protein was strongly downregulated in the combo, suggesting protein degradation (Figure 4B,E). A more detailed time course analysis showed that, while the transcript tended to increase over time under combined treatment, the protein decreased sharply and then partially recovered but remained below initial levels (Appendix A). Altogether, these results suggest that chemotherapy may cooperate with ALK inhibition in ALCL to block cell proliferation and promote cell death.

Crizotinib plus decitabine induced the modulation of several interesting targets as compared to single agents (Figure 5 and Appendix A). This combination caused a significant downregulation of the DNA methyltransferase 1 gene (DNMT1; Figure 5A,D), which correlated with an increase in histone H3 methylation on lysine 4 (H3K4me, a marker of active transcription; Figure 5G) and higher expression of the transcription factor GATA3, which is epigenetically silenced in ALK+ ALCL compared to normal T cells [17]. Furthermore, consistent with a reduction in DNMT1 expression, we observed an upregulation of STAT5A, RAB13, GATA3, TGFB1 and IL-2RG (Figure 5A,D), which are all DNMT1 target genes, epigenetically silenced in ALK+ ALCL [17,18,20,29]. A strong reduction in NPM/ALK downstream signaling was observed, as shown by reduction in ERK1/2, p38-MAPK and JNK phosphorylation (Figure 5G). Interestingly, decitabine-resistant cells selected under long-term exposure to the drug showed increased DNMT1 transcription (Appendix A). In both cell lines, the synergic induction of cell death was associated with a strong upregulation of pro-apoptotic genes FAS, NOXA, BAK and BIM (Figure 5B,E). Induction of cell cycle inhibitors was observed in combination-treated cells (Figure 5C,F), in line with the reduction in DNMT1 and the increase in TGFβ1, both of which are known to control CDKN2A (p14ARF and p16INK4A) and CDKN1A (p21Waf1) transcription [20,30,31]. A discrepancy was again noted between CDKN1A mRNA and p21Waf1 protein levels in the presence of the combined treatment, suggesting degradation of the cell cycle inhibitor protein (Figure 5C,F,G). Expression of p53 protein was induced by the combination (Figure 5G). An interesting complete suppression of CD30 expression was seen only in Karpas-299 cells following exposure to combined crizotinib+decitabine (Appendix A). The biological meaning of this effect is under investigation. Finally, as some microRNAs have been shown to have a role in ALCL [32,33], we analyzed their expression. A small but significant increase in miR-939 was detected only in Karpas-299 cells (Appendix A). These results indicate that adding a DNMT1 inhibitor to an ALK inhibitor may achieve deeper blockade of NPM/ALK-driven oncogenic signaling and enhance cell death in ALK+ ALCL cells.

The activation of ALK downstream pathways was analyzed by Western blot following crizotinib+trametinib treatments (Figure 6A). As expected, ERK1/2 phosphorylation was efficiently inhibited by trametinib, while crizotinib only caused a slight decrease in p-ERK signal compared to control. On the other hand, crizotinib, but not trametinib, completely blocked STAT3 activation. In contrast, both pathways were effectively suppressed in cells under combined treatment (Figure 6A). Therefore, crizotinib and trametinib cooperate to achieve a more profound impairment of NPM/ALK signal transduction, thus blocking the main pathogenic mechanisms leading to cell survival and proliferation. Consequently, these two drugs acted synergistically to trigger apoptosis, as demonstrated by reduction in survivin [34] and increase in BIM protein expression (Figure 6A). Gene expression analysis confirmed the induction of pro-apoptotic genes (Figure 6B,C and Appendix A). Interestingly, the combination reversed crizotinib-induced upregulation of pro-survival genes BCL2 and BCL-XL that have been shown to hamper crizotinib efficacy in ALCL [35]. Once again, p21 protein was downregulated by the combination (Figure 6A). Finally, as previously noted with lorlatinib [9], crizotinib-resistant cells showed slightly higher ERK activation, suggesting that preemptive blockade of MEK by trametinib precluded the use of this bypass mechanism in the combo (Appendix A). These data suggest that trametinib cooperates with ALK inhibitors to achieve broader and deeper inhibition of the oncogenic program generated by the fusion kinase and impedes the off-target reactivation of MAPK signaling.

## 4. Discussion

About 30% of ALK+ ALCL patients relapse or are refractory to first-line CHOP-based therapy and approximately half of these patients develop further resistance to ALK inhibitors [3]. This work explored the possibility to prevent, rather than overcome, resistance. We demonstrate that this goal can be achieved using upfront rational combinations to simultaneously block more than one pathway supporting tumor progression, thus decreasing the probability to develop resistance. Guided by previous evidence on the therapeutic impact of a drug combination approach in ALK+ tumors [9,12,23,36,37,38], we investigated the potentialities of upfront combined treatments by testing four distinct drug combinations on different ALK+ ALCL cell lines. Our results show that crizotinib combined with decitabine or chemotherapy or trametinib impairs the rise of resistant clones in the long term, in contrast to single agents. Statistical analysis supported these findings demonstrating that these combinations have synergistic effects, explaining the efficacy of the combined treatments. Using different in vitro and in vivo assays, we showed that these drug combinations have the potential to improve current protocols based on sequential monotherapies.

By contrast, combining ALK inhibitors with each other was unsuccessful in most cases. The failure of this strategy may be due to the activity profiles of the inhibitors that are not perfectly complementary or to the occurrence of ALK-independent resistance. In the latter case, combining an ALK inhibitor with other agents would be more beneficial. We therefore focused our attention on combinations of drugs hitting different targets, i.e., crizotinib+decitabine, crizotinib+CHO and crizotinib+trametinib. We found that all these combinations efficiently inhibited the activation of ALK downstream signaling by abolishing the phosphorylation of effector molecules, such as STAT3, ERK, JNK and p38-MAPK. This eventually resulted in a synergistic reduction in tumor cell growth and promotion of cell death, and most importantly, long-term suppression of resistance. It is worth noting that the molecular mechanisms supporting improved efficacy of combinations were similar in two different cell lines, proving the validity of the data and the irrelevance of the cellular genetic background in determining the final effect of a combination. The three combined treatments were found to induce cell cycle arrest and apoptosis by upregulating cell cycle inhibitors and pro-apoptotic genes, and through modulation of survival factors. All three combinations enforced BIM expression, confirming its central role in ALCL cells [19]. Counterbalancing of crizotinib-induced BCL2 upregulation likely contributed to the increased efficacy of the drug combinations. In general, while some genes were individually modulated by single agents as well as by the combination, the latter induced a broader effect by regulating more than one transcript; it is the case of the CDKN2A locus in Karpas-299 cells with crizotinib+decitabine, or STAT3 and ERK activation with crizotinib+trametinib. The epigenetic combo promoted several additional changes, including downregulation of DNMT1 and increase in TGFB1, both of which indirectly affect cell cycle progression [20,30]. Interestingly, restoration of STAT5A and IL2RG expression has been reported to impair NPM/ALK expression in ALCL cells, thus acting as tumor suppressors in this context [18,29]. Their upregulation by crizotinib+decitabine combination may account, at least in part, for the observed inactivation of ALK downstream signaling. Furthermore, IL2RG has been found overexpressed in patients with low NPM/ALK expression and less aggressive disease [39].

Finally, an intriguing common theme among different combinations was the discrepancy between mRNA and protein levels of p21Waf1/Cip1. A similar response has been described in several cell types after UV irradiation [40]. Unusual roles of p21 protein, usually cited as a cell cycle inhibitor, have been previously demonstrated. In particular, p21 has been shown to protect from cell death after acute insults [41,42]; its immediate degradation would therefore lead to death.

Combining a selective drug (ALK inhibitor) with an unspecific anticancer agent which allows blind killing of TKI-resistant clones may prevent drug resistance. This approach has been explored in other contexts [43,44]. In particular, combining an epigenetic drug with targeted drugs has been postulated to hold superior potential due to the ability to block epigenetic events that drive cancer stem cell survival and feed resistance [45,46]. Synergism should not only lead to enhanced efficacy, but also to a reduction in toxicity due to the use of reduced drug doses. However, this issue needs more investigation in order to establish safe drug combinations. Recently, early combinations of TKIs with chemotherapy improved long-term survival of acute lymphoblastic leukemia patients, but suggested increased toxicity [47,48]. When we tested the efficacy of these combinations in a murine model, crizotinib+decitabine significantly delayed tumor growth; however, signs of toxicity were observed in some of the mice treated with decitabine, both alone and in combination. This was likely related to the dosage of the epigenetic drug administered. Therefore, the problem of toxicity could be overcome by fine tuning drug dosing to achieve important antitumor effects and a tolerable toxicity profile at the same time. Further in vivo studies are in progress to find a balance between efficacy and toxicity, as well as to determine the long-term outcome of these combinations in mice. While extensive investigation on toxicity can be run in animals, caution will have to be used in human subjects; maximum tolerated doses of combinations will need to be established in clinical trials. On the other hand, the combination of two targeted therapies (crizotinib and trametinib) allowed safe and effective tumor growth inhibition in mice. The impact of MAPK pathway in ALK+ cancer has been suggested by various studies [9,12,14,49]. Indeed, trials of ALK inhibitor plus MEK inhibitor are underway in ALK+ NSCLC [50].

## 5. Conclusions

In conclusion, here we provide proof of principle for a combined selective treatment to restrain TKI resistance in ALCL models. These data collectively demonstrate that an upfront combination of targeted and cytotoxic therapies might be beneficial in the treatment of ALK+ ALCL. This is further corroborated by recent results of brentuximab+chemotherapy combinations in newly diagnosed ALK+ ALCL patients [4,51] and promising in vitro data on crizotinib+brentuximab association [52]. Moreover, the meaning of this work extends to a wider scope since the combinatorial approach represents a relevant strategy in all clinical fields for which resistance represents an urgent problem to be solved.

## Figures and Tables

**Figure 1 cancers-13-04422-f001:**
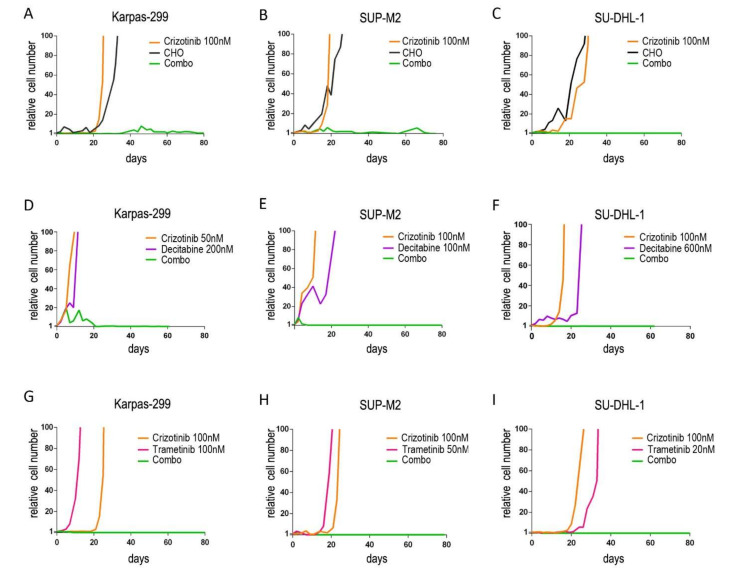
Long-term expansion of drug-resistant cells under continuous treatment. NPM/ALK+ ALCL cells (**A**,**D**,**G**): Karpas-299; (**B**,**E**,**H**): SUP-M2; (**C**,**F**,**I**): SU-DHL-1 were cultured in the presence of the indicated drugs for up to 100 days. Cumulative cell count, relative to day 0, is shown. CHO = 2 µM cyclophosphamide, 20 nM doxorubicin, 0.5 nM vincristine. Combo = combination. Decitabine and trametinib concentration varies among cell lines, accounting for different sensitivities.

**Figure 2 cancers-13-04422-f002:**
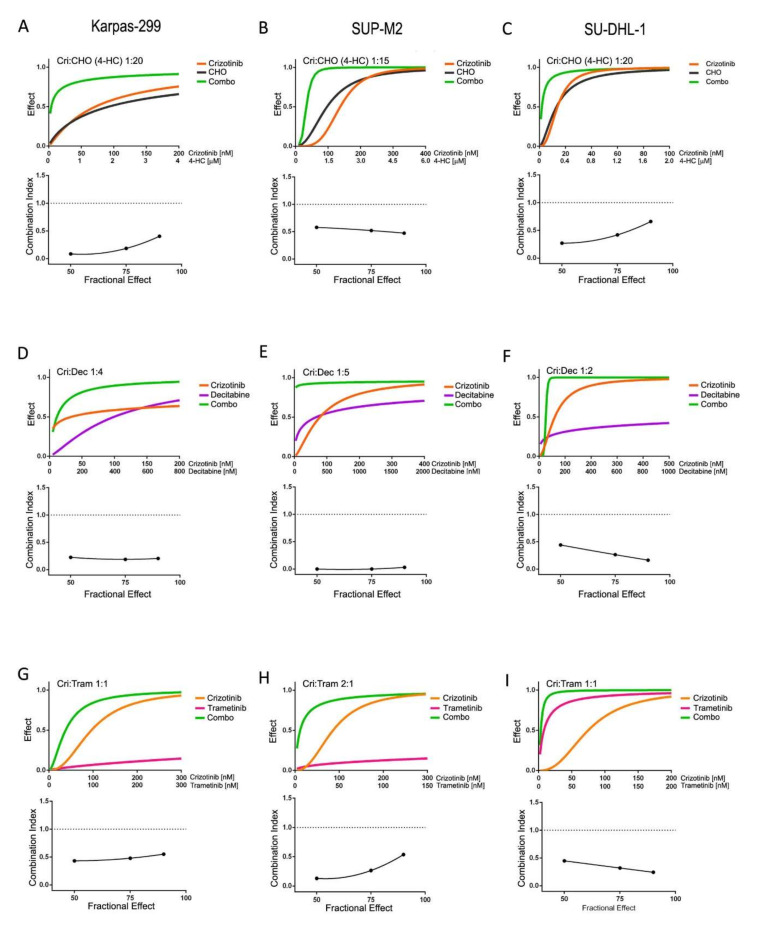
Synergistic drug combinations in ALK+ cells in short-term cultures. (**A**,**D**,**G**): Karpas-299; (**B**,**E**,**H**): SUP-M2; (**C**,**F**,**I**): SU-DHL-1; cells were cultured for 3 days in the presence of inhibitors as single agents or in constant ratio combinations or vehicle alone. MTS assay was used to assess cell culture growth and viability. In each panel, the top graph (dose–effect curve) shows the fractional effect as a function of drug concentrations (0 = no effect; 1 = complete inhibition); the bottom graph reports combination indexes as a function of the fraction affected according to Chou and Talalay [22]. A dotted line at C.I. = 1 highlights the level of an additive effect. For CHO treatments, dosing of 4-HC and crizotinib:4-HC drug ratios are reported, while vincristine and doxorubicin were added proportionally, keeping a constant ratio to 4-HC (see Section 2). Combo = combination.

**Figure 3 cancers-13-04422-f003:**
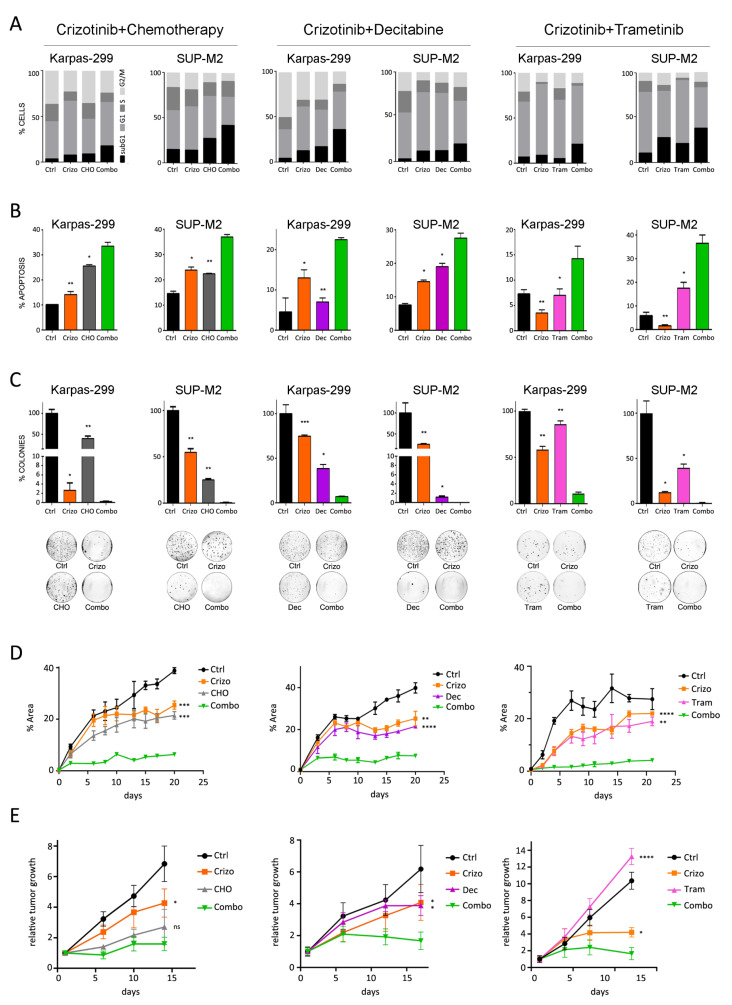
Biological characterization of the synergism. Karpas-299 and SUP-M2 cells were treated with single drugs or with combinations of crizotinib+CHO (left), crizotinib+decitabine (middle) and crizotinib+trametinib (right). Drug concentrations were as in Figure 1. Combo = combination. (**A**) Cell cycle was analyzed by propidium iodide staining. The percentage of cells in each phase of the cell cycle is reported. (**B**) The percentage of apoptotic cells was assessed by annexin V staining. (**C**) Cells were seeded in medium containing agar in the presence of drugs. Colonies were counted after 21 days, and the number is reported as percent of vehicle-treated control. Photographs from a representative experiment replicate are shown below the graph. (**D**) SUP-M2 cells were embedded in Matrigel, seeded in 96-well plates and left to grow for 96 h. Then the indicated treatments were started (day 0). The data report, at each time point, the total area occupied by spheroid masses in the wells, calculated by ImageJ software. (**E**) Karpas-299 cells were inoculated s.c. in the flank of SCID mice. After 10 days, crizotinib (30 mg/kg), decitabine (0.8 mg/kg), trametinib (2 mg/kg) and CHO treatments were started as described in Section 2. Values shown are tumor volumes relative to day 1 of dosing (mean ± SD). In all panels, asterisks indicate statistical significance versus the combination (two-tailed *t*-test; *, *p* < 0.05; **, *p* < 0.01; ***, *p* < 0.001; ****, *p* < 0.0001).

**Figure 4 cancers-13-04422-f004:**
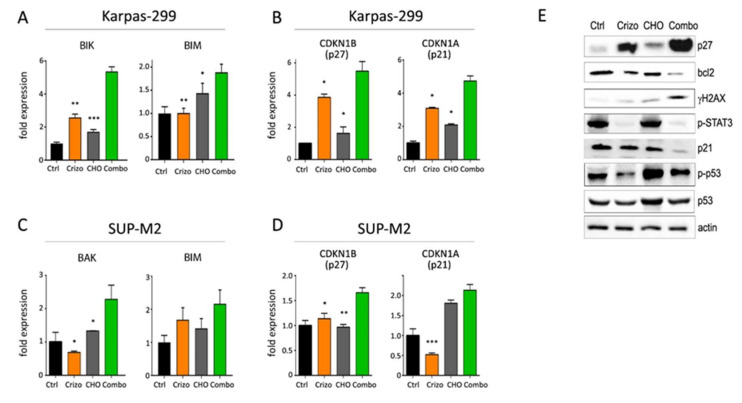
Molecular effects of crizotinib+CHO combination. (**A**–**D**) Quantitative real-time PCR analysis of selected genes involved in apoptosis (**A**,**C**) and cell cycle regulation (**B**,**D**) is shown in Karpas-299 (**A**,**B**) and SUP-M2 (**C**,**D**) cells treated for 72 h with the indicated drugs, using the same concentrations shown in Figure 1. (**E**) Western blot analysis of Karpas-299 cells treated with single or combined crizotinib+CHO for 72 h, uncropped Western blot figure in Appendix A. Ctrl = control; Crizo = crizotinib; Combo = combination. Primary antibodies employed in this study are listed in Appendix A. Primers were described previously [19,24] or are listed in Appendix A. Asterisks indicate statistical significance vs. combo. *, *p* < 0.05; **, *p* < 0.01; ***, *p* < 0.001.

**Figure 5 cancers-13-04422-f005:**
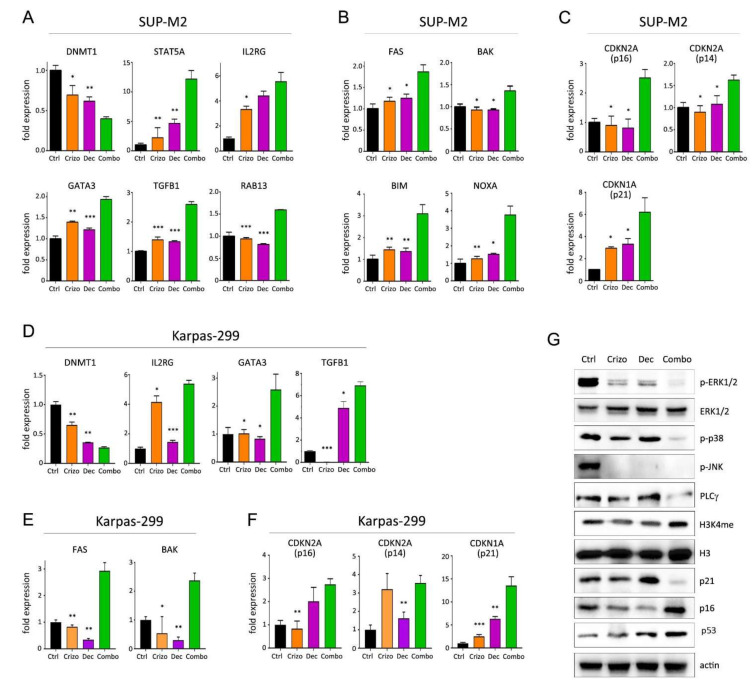
Molecular effects of crizotinib+decitabine combination. (**A**–**F**) Quantitative real-time PCR analysis of epigenetically regulated genes (**A**,**D**) and genes involved in apoptosis (**B**,**E**) and cell cycle regulation (**C**,**F**), in SUP-M2 (**A**–**C**) and Karpas-299 (**D**–**F**) cells treated for 72 h with the indicated drugs. (**G**) Western blot analysis of SUP-M2 cells treated with single or combined crizotinib+decitabine for 72 h, uncropped Western blot figure in Appendix A. Ctrl = control; Crizo = crizotinib; Dec = decitabine; Combo = combination. The same drug concentrations shown in Figure 1 were used. Asterisks indicate statistical significance vs. combo. *, *p* < 0.05; **, *p* < 0.01; ***, *p* < 0.001.

**Figure 6 cancers-13-04422-f006:**
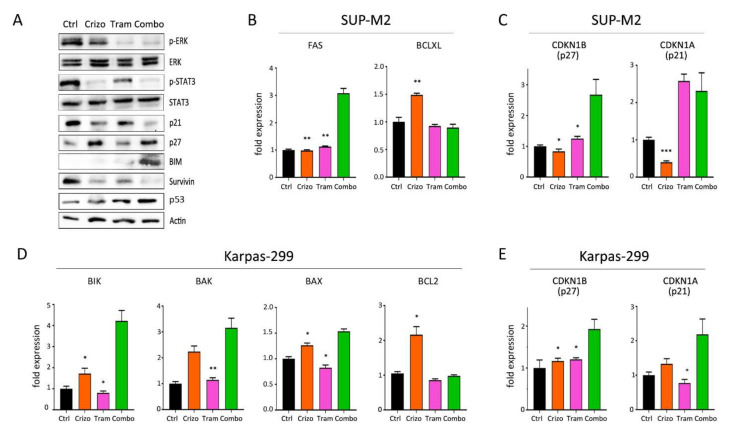
Molecular effects of crizotinib+trametinib combination. (**A**) Western blot analysis of Karpas-299 cells treated with single or combined crizotinib+trametinib for 120 h, uncropped Western blot figure in Appendix A. (**B**–**E**) Quantitative real-time PCR analysis of selected genes involved in apoptosis and cell cycle regulation in SUP-M2 (**B**,**C**) and Karpas-299 (**D****,E**) cells. Ctrl = control; Crizo = crizotinib; Tram = trametinib; Combo = combination. The same drug concentrations shown in Figure 1 were used. Asterisks indicate statistical significance vs. combo. *, *p* < 0.05; **, *p* < 0.01; ***, *p* < 0.001.

**Table 1 cancers-13-04422-t001:** Combination index values obtained at ED50, ED75 and ED90 from cell proliferation experiments with crizotinib:decitabine (CRI:DEC), crizotinib:chemotherapy (CRI:CHO) and crizotinib:trametinib (CRI:TRAM) combinations in three ALK+ cell lines and one ALK-negative cell line (U937). The level of synergism is calculated on the mean of the three values according to Chou and Talalay [22]. Synergy, additivity and antagonism are indicated by green, yellow and red color, respectively, in the right-most column. Results are the average of at least three independent experiments.

Combination	Cell Line	Ratio	EC50	EC75	EC90	Mean	Level of Synergism	
**CRI:DEC**	Karpas 299	1:1	0.000	0.006	0.007	0.004	Very Strong Synergism	
	Karpas 299	1:4	0.226	0.189	0.206	0.207	Strong Synergism	
	Karpas 299	1:20	0.509	0.512	0.560	0.527	Synergism	
	Karpas 299	1:100	0.142	0.336	0.811	0.429	Synergism	
	SU-DHL-1	5:1	0.448	0.301	0.203	0.317	Synergism	
	SU-DHL-1	1:1	0.644	0.632	0.631	0.636	Synergism	
	SU-DHL-1	1:2	0.442	0.265	0.165	0.291	Strong Synergism	
	SU-DHL-1	1:5	0.779	0.675	0.641	0.698	Synergism	
	SUP-M2	5:1	1.080	0.713	0.481	0.758	Moderate Synergism	
	SUP-M2	1:1	0.711	0.839	1.110	0.887	Slight Synergism	
	SUP-M2	1:2	0.536	0.391	0.351	0.426	Synergism	
	SUP-M2	1:5	0.000	0.001	0.038	0.013	Very Strong Synergism	
	U937	1:1	>10	>10	>10	>10	Very Strong Antagonism	
	U937	1:2	>10	>10	>10	>10	Very Strong Antagonism	
**CRI:CHO**	Karpas 299	1:22	0.087	0.185	0.405	0.226	Strong Synergism	
	Karpas 299	1:50	0.426	0.545	0.698	0.556	Synergism	
	Karpas 299	1:66	0.432	0.346	0.284	0.354	Synergism	
	Karpas 299	1:150	0.500	0.625	0.781	0.635	Synergism	
	SU-DHL-1	1:40	0.236	0.359	0.555	0.383	Synergism	
	SU-DHL-1	1:20	0.268	0.418	0.662	0.449	Synergism	
	SU-DHL-1	1:10	1.298	1.420	1.571	1.430	Moderate Antagonism	
	SUP-M2	1:16	0.579	0.521	0.473	0.524	Synergism	
	SUP-M2	1:32	0.845	0.715	0.611	0.724	Moderate Synergism	
	SUP-M2	1:64	0.869	0.759	0.667	0.765	Moderate Synergism	
	U937	1:40	1.015	0.915	0.830	0.920	Nearly Additive	
	U937	1:20	1.096	1.044	1.008	1.050	Nearly Additive	
	U937	1:10	1.207	1.287	1.407	1.300	Moderate Antagonism	
**CRI:TRAM**	Karpas 299	1:1	0.616	0.589	0.563	0.589	Synergism	
	Karpas 299	1:2	0.510	0.477	0.448	0.478	Synergism	
	Karpas 299	1:4	0.228	0.336	0.538	0.367	Synergism	
	Karpas 299	1:3	0.433	0.480	0.553	0.489	Synergism	
	SU-DHL-1	1:9	0.967	0.665	0.461	0.698	Synergism	
	SU-DHL-1	1:3	0.758	0.572	0.438	0.589	Synergism	
	SU-DHL-1	1:1	0.450	0.321	0.244	0.338	Synergism	
	SU-DHL-1	3:1	0.046	0.053	0.068	0.056	Very Strong Synergism	
	SUP-M2	1:2	0.483	0.542	0.615	0.546	Synergism	
	SUP-M2	1:1	0.579	0.599	0.624	0.600	Synergism	
	SUP-M2	2:1	0.132	0.266	0.538	0.312	Synergism	
	U937	1:1	>10	>10	>10	>10	Very Strong Antagonism	
	U937	1:3	>10	>10	>10	>10	Very Strong Antagonism	

## Data Availability

The data presented in this study are available in the article and in Appendix A.

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
