# Peer review of "Synergistic Drug Combinations Prevent Resistance in ALK+ Anaplastic Large Cell Lymphoma"

_cancers, 2021, doi:10.3390/cancers13174422_

Round 1
Reviewer 1 Report
In the article by Giulia Arosio et al, the authors investigated the efficacy of rational drug combinations to prevent the rise of resistant ALK+ ALCL, in vitro and in vivo.
Since ALK+ ALCL is a rare lymphoma, preclinical evaluation of drug efficacy is important, in order to best select the drugs of interest to be evaluated in clinical trials.
The paper is well written and easy to read, the methodology is good, and results are convincing. I have only minor comments.
1) Title
Please add « ALK+ » ALCL
2) Introduction
- « ALK+ ALCL, a non-Hodgkin peripheral T-cell lymphoma that depends on the activity of the NPM/ALK chimeric protein »: NPM is not always the ALK partner gene.
- Since the current treatment of adult ALK+ ALCL is mainly BV-CHP or CHOEP, these regimens should be written (instead of « CHOP-like ») and the 2 main studies should be cited : Horwitz S et al: Brentuximab vedotin with chemotherapy for CD30-positive peripheral T-cell lymphoma (ECHELON-2): a global, double-blind, randomised, phase 3 trial, Lancet 2019. And Sibon D et al: ALK-positive anaplastic large-cell lymphoma in adults: an individual patient data pooled analysis of 263 patients, Haematologica 2019.
3) Results
a. Figure 1: why is the crizotinib concentration 50 nM in Fig-D, when it is 100 nM in all other cases? Same for decitabine and trametinib, the concentrations of which are different between experiments. It is stated that « Preliminary tests were run to determine drug concentrations that would allow the outgrowth of resistant cells as single agents (not shown) », but crizotinib is at different concentrations for the same cell line (Karpas-299). This should be explained more clearly in the text. b. Figure 1, S3, S5: the scale of the x-axis should be the same for all figures (for example, Fig-D and Fig-F end at 60 instead of 80).
Author Response
1) Title
« ALK+ » ALCL was added to the title as suggested by the Reviewer
2) Introduction
We added the information and the references as suggested
3) Results
- the text was modified to better explain this point
- the scale of the x-axis was made the same for all panels in figures 1 and S3. Legend to Fig.S5 now explains why the axis differs among the various experiments.

Reviewer 2 Report
This study looked at how the combination with ALK inhibitor crizotinib and other agents can improve the activity to ALK+ALCL cell line.
The result of this study is beneficial to consider next clinical trial in this patient population.
Major comments:
- How did you pick the combination partner? Clinically, we have brentuximab + CHOP as established first line treatment based on ECHELON-2 so not sure if crizotinib + CHOP is important. Have you tested or thought for crizotinib + brentuximab, jak2 inhibitor or BCL-2 inhibitor?
- How do you explain long term responders to critizotinib on human trial (retreatment also works) if cell line model shows universal resistance in short period of time?
- I would recommend extending X axis on figure 1 to show when cell lines become resistant to combinations
- I suggest revising introduction and discussion slightly to reflect what we know for ALK+ALCL. For example, do we really know “subpopulation” exist in ALCL? How can we overcome the problem of toxicity, what is involved in the “fine tuning”? We don’t design trial based on this, we only use MTD by looking at DLT. How can we do this in human beings?
Overall, I think this is important pre-clinical study.
Author Response
- We agree that Brentuximab + CHOP is an important advance in the field and corroborates the idea of using rational combinations as standard therapies. A comment on this point was added in the Conclusion. We indeed planned to run crizotinib + brentuximab combo as well, but had difficulties to obtain BV. Moreover, while this study was underway, this combination was tested and shown to be synergistic by Hudson et al. (Pediatr Blood Cancer 2018). This reference has been added.
- We are well aware of long responders to crizotinib, however we wanted to model resistant cases. To do this we used suboptimal drug doses to see if in this cases a combination would perform better than single drugs. Manuscript text has been modified to clarify this point, as suggested by the Reviewer.
- The experiments shown in figure 1 were kept running up to 100 days and no resistant clone was observed in the combinations.
- Introduction and Discussion have been revised as suggested.

Reviewer 3 Report
In their manuscript, Arosio and colleagues proposed an approach of a combination of different drugs as a treatment for ALCL as a frontline therapy to prevent relapse. They propose reasonable treatment combinations based on previous findings of their own and other groups. A highlight lies on the exploration of resistance to ALK-inhibitors. Further, they propose a mechanistical explanation for the synergism of certain treatment combination. This findings are in general very interesting and might be a base for further studies. However, some improvisations and experiments will make the data more robust and should be conducted.
- The authors present ALK+ ALCL as an adult tumor entity. However, it is actually in 80% of the cases a pediatric disease, especially for ALK translocated ALCL.
- CHOP is not the standard treatment for pediatric ALCL in Europe. This has to be at least mentioned and its chemotherapeutics considered for this study.
- The authors present the concept of combination therapy as a frontline cancer treatment to prevent therapy resistance as novelty. However, this approach is in general widely used for different cancers in clinical treatment protocols. Already in the case of ALK+ ALCL it is used e.g. by combining different chemotherapeutic drugs with each other and further e.g. with Brentuximab.
- How were the doses of the chemotherapeutics for the experiments determined? It is stated that this data is not shown, however this information is important. For example 20nM of Doxorubicin to treat ALCL cells is a rather low dose that is most likely sublethal. Therefore it is not surprising that resistant cells arise after such a treatment. In our hands an IC50 dose was around 100nM. The IC50 of all used drugs for the selected cell lines in this study has to be determined and it has to be justified why which dose was used, e.g. treatment with IC50 dose, or sublethal treatment with IC10 etc.
- The authors state that Crizotinib resistance goes in line with an upregulation of e.g. the Map kinase pathway and that is why the authors chose e.g. trametinib. It has to be shown that this pathway is really upregulated in the used cell lines, for example by Western blotting. The same applies for the epigenetic treatment.
- All drug treatments should be performed as a control in an ALK neg. ALCL cell line. This is especially informative for the ALK-inhibitors.
- Since all the treatment combination include an ALK-inhibitor and center around ALK-inhibitor resistance, it should be explored, whether those combination treatments are effective in other ALK-alterated cancers, such as ALK-mutated neuroblastoma or non-small-cell lung cancer.
- The in vivo experiment should be performed longer. Do in the combination treatment setting resistant cells also arise after a certain time?
- The mechanism of synergism should be further analyzed e.g. by RNA sequencing of treated cells and detection of enrichment of for example apoptosis-relevant pathways or gene sets.
- The discussion only briefly mentions combinatorial treatments in other cancers. This has to be extended. Especially ALK-alterated cancers have to be mentioned. What other studies have been conducted in ALCL? What are upcoming clinical trials?
Author Response
- The text has been amended following the Reviewer’s suggestions, for what pertains clinical standard regimens for adult and paediatric ALCL, brentuximab combinations with chemotherapy, and other experimental drug combinations currently under investigation.
- The text was also modified to better clarify some points like the choice of drug concentrations used in our study.
- We have previously shown the involvement of MAPK pathway in drug resistance (ref. 9); we have now added a comment on this. As suggested by the Reviewer, Western blot was run to compare MAPK activity in parental versus crizotinib-resistant cells obtained in this work. The results have been added into the Supplementary data.
- The lack of activity of all combos in an ALK-negative cell line (U937) is shown in Table 1. To further show this in ALCL cells, as requested by the Reviewer, we analyzed the sensitivity to crizotinib of MAC1 cells. The results were added in Supplementary data.
- Additional experiments, like extending to other ALK-addicted tumors, performing longer in vivo trials and comprehensive transcriptomics analyses are the focus of an ongoing project that will be reported as a follow up to this initial proof-of-concept paper.
